# ADVERSARIAL META-LEARNING

## ABSTRACT

Meta-learning enables a model to learn from very limited data to undertake a new task. In this paper, we study the general meta-learning with adversarial samples. We present a meta-learning algorithm, ADML (ADversarial Meta-Learner), which leverages clean and adversarial samples to optimize the initialization of a learning model in an adversarial manner. ADML leads to the following desirable properties: 1) it turns out to be very effective even in the cases with only clean samples; 2) it is robust to adversarial samples, i.e., unlike other meta-learning algorithms, it only leads to a minor performance degradation when there are adversarial samples; 3) it sheds light on tackling the cases with limited and even contaminated samples. It has been shown by extensive experimental results that ADML outperforms several representative meta-learning algorithms in the cases involving adversarial samples generated by different attack mechanisms, on two widely-used image datasets, MiniImageNet and CIFAR100, in terms of both accuracy and robustness.

## 1 INTRODUCTION

Deep learning has made tremendous successes and emerged as a *de facto* approach in many application domains, such as computer vision and natural language processing, which, however, depends heavily on huge amounts of labeled training data. The goal of meta-learning is to enable a model (especially a Deep Neural Network (DNN)) to learn from only a small number of data samples to undertake a new task, which is critically important to machine intelligence but turns out to be very challenging. Currently, a common approach to learn is to train a model to undertake a task from scratch without making use of any previous experience. Specifically, a model is initialized randomly and then updated slowly using gradient descent with a large number of training samples. This kind of time-consuming and data-hungry training process is quite different from the way how a human learns quickly from only a few samples and obviously cannot meet the requirement of meta-learning. Several methods (Finn et al. (2017); Vinyals et al. (2016); Snell et al. (2017); Sung et al. (2018)) have been proposed to address meta-learning by fixing the above issue. *For example, a well-known work (Finn et al. (2017)) presents a novel meta-learning algorithm called MAML (Model-Agnostic Meta-Learning), which trains and optimizes the initialization of model parameters carefully such that it achieves the maximal performance on a new task after its parameters are updated through one or just a few gradient steps with a small amount of data.* This method is claimed to be model-agnostic since it can be directly applied to any learning model that can be trained with gradient descent.

*Robustness is another major concern for machine intelligence, especially for the safety-critical applications, such as facial recognition, algorithmic trading and copyright control. It has been shown that such learning models can be easily fooled by adversarial manipulation to cause serious security threats (Zhao et al. (2018); Goldblum et al. (2020); Saadatpanah et al. (2019)), which, however, can be properly and effectively handled by conventional adversarial training and pre-processing defenses (Madry et al. (2017); Zhang et al. (2019); Samangouei et al. (2018)). Nonetheless, if the data is limited (e.g., face recognition from few images), the aforementioned pipelines, which require a large amount of training data, suffers from serious performance degradation. Although meta-learning based approaches show great potential on dealing with few-shot tasks, we show via experiments that existing meta-leaning algorithms (such as MAML (Finn et al. (2017)), Matching Networks (Vinyals et al. (2016)) and Relation Networks (Sung et al. (2018))) are also vulnerable to adversarial samples, i.e., adversarial samples can lead to a significant performance degradation*

*for meta-learning.* To the best of our knowledge, existing works on meta-learning have not yet addressed adversarial samples, which, however, is the main focus of this paper.

In this paper, we extend meta-learning to a whole new dimension by studying how to quickly train a model (especially a DNN) for a new task using a small dataset with both clean and adversarial samples. Since both meta-learning and adversarial learning have been studied recently, a straightforward solution is to simply combine MAML (Finn et al. (2017)) algorithm with adversarial training (e.g., Goodfellow et al. (2015)). However, we show such an approach does not work well by our experimental results. We present a novel ADversarial Meta-Learner (ADML), which utilizes antagonistic correlations between clean and adversarial samples to let the inner gradient update arm-wrestle with the meta-update to obtain a good and robust initialization of model parameters. Hence, "adversarial" in ADML refers to not only adversarial samples but also the way of updating the learning model. The design of ADML leads to several desirable properties. First, it turns out to be very effective even in the cases with only clean samples. Second, unlike other meta-learning algorithms, ADML is robust to adversarial samples since it only suffers from a minor performance degradation when encountering adversarial samples, and it outperforms several representative meta-learning algorithms (Finn et al. (2017); Vinyals et al. (2016); Sung et al. (2018); Bertinetto et al. (2019)) in such cases. In addition, ADML is agnostic to the attack mechanism, which is responsible for the adversarial samples generation. Most importantly, it opens up an interesting research direction and sheds light on dealing with the cases with limited and even contaminated samples, which are common in real life. We conducted a comprehensive empirical study for performance evaluation using two widely-used image datasets, MiniImageNet (Vinyals et al. (2016)) and CIFAR100 (Krizhevsky et al. (2009)). Experimental results well justify the effectiveness and superiority of ADML in terms of both accuracy and robustness.

## 2 RELATED WORK

**Meta-Learning:** Research on meta-learning has a long history, which can be traced back to some early works (Naik & Mammone (1992); Thrun & Pratt (1998)). Meta-learning, a standard methodology to tackle few-shot learning problems, has recently attracted extensive attention due to its important roles in achieving human-level intelligence. Several specialized models (Vinyals et al. (2016); Koch et al. (2015); Snell et al. (2017); Sung et al. (2018)) have been proposed for meta-learning, particularly for few-shot classification, by comparing similarity among data samples. Specifically, Koch et al. (2015) leveraged a Siamese Networks to rank similarity between input samples and predict if two samples belong to the same class. In addition, Relation Networks (Sung et al. (2018)) was proposed to classify query images by computing relation scores, which can be extended to few-shot learning. Vinyals et al. (2016) presented a neural network model, Matching Networks, which learn an embedding function and use the cosine distance in an attention kernel to measure similarity.

Another popular approach to meta-learning is to develop a meta-learner to optimize key hyperparameters (e.g., initialization) of the learning model. Specifically, Finn et al. (2017) presented a model-agnostic meta-learner, MAML, to optimize the initialization of a learning model with the objective of maximizing its performance on a new task after updating its parameters with a small number of samples. Several other methods (Andrychowicz et al. (2016); Ravi & Larochelle (2017); Santoro et al. (2016); Mishra et al. (2018)) utilized an additional neural network, such as LSTM, to serve as the meta-learner. A seminal work (Andrychowicz et al. (2016)) developed a meta-learner based on LSTMs and showed how the design of an optimization algorithm can be cast as a learning problem. Ravi & Larochelle (2017) proposed another LSTM-based meta-learner to learn a proper parameter update and a general initialization for the learning model. A recent work (Mishra et al. (2018)) presented a class of simple and generic meta-learners that use a novel combination of temporal convolutions and soft attention.

**Adversarial Learning:** DNN models have been shown to be vulnerable to adversarial samples. Particularly, Szegedy et al. (2014) showed that they can cause a DNN to misclassify an image by applying a certain hardly perceptible perturbation, and moreover, the same perturbation can cause a different network (trained on a different subset of the dataset) to misclassify the same input. It has also been shown by Goodfellow et al. (2015) that injecting adversarial samples during training can increase the robustness of DNN models. Papernot et al. (2017) introduced the first practical demonstration of a black-box attack controlling a remotely hosted DNN without either the model

internals or its training data. In addition, Kurakin et al. (2017) studied adversarial learning at scale by proposing an algorithm to train a large scale model, Inception v3, on the ImageNet dataset, which has been shown to significantly increase the robustness against adversarial samples.

To the best of our knowledge, meta-learning has not been well studied in the setting with adversarial samples. We not only show a straightforward solution does not work well but also present a novel and effective method, ADML.

## 3 ADVERSARIAL META-LEARNING

### 3.1 PROBLEM STATEMENT

The regular machine learning problem seeks a model that maps observations $\mathbf{x}$ to output $\mathbf{y}$; and a training algorithm optimizes the parameters of the model with a training dataset, whose generalization is then evaluated on a test dataset. While in the setting of meta-learning, the learning model is expected to be trained with limited data to be able to adapt to a new task quickly. Meta-learning includes meta-training and meta-testing. In the *meta-training* phase, we use a set $\mathcal{T}$ of $T$ tasks, each of which has a loss function $\mathcal{L}_i$, and a dataset $\mathcal{D}_i$ (with limited data) that is further split into $\mathbf{D}_i$ and $\mathbf{D}_i'$ for training and testing respectively. For example, in our experiments, each task is a 5-way classification task. *Note that the 5-way represents 5 different classes.*

We aim to develop a meta-learner (i.e., a learning algorithm) that takes as input the datasets $\mathcal{D} = \{\mathcal{D}_1, \cdots, \mathcal{D}_T\}$ and returns a model with parameters $\boldsymbol{\theta}$ that maximizes the average classification accuracy on the corresponding testing sets $\mathcal{D}' = \{\mathbf{D}_1', \cdots, \mathbf{D}_T'\}$. Note that here these testing data are also used for meta-training. In the *meta-testing*, we evaluate the generalization of the learned model with parameters $\boldsymbol{\theta}$ on new tasks, whose corresponding training and test datasets may include adversarial samples. The learned model is expected to learn quickly from just one (1-shot) or $K$ ($K$-shot) training samples of a new task and deliver highly-accurate results on its testing samples. An ideal meta-learner is supposed to return a learning model that can deal with new tasks with only clean samples; and suffers from only a minor performance degradation for new tasks with adversarial samples. We believe the proposed ADML can be easily extended to other scenarios as long as adversarial samples can be properly generated.

### 3.2 ADVERSARIAL META-LEARNER (ADML)

We formally present the proposed ADML as Algorithm 1 for *meta-training*. We consider a model $f_{\boldsymbol{\theta}}$ parameterized by $\boldsymbol{\theta}$, which is updated iteratively. Here, an updating *episode* includes an inner gradient update process (Line 8–Line 12) and a meta-update process (Line 14). Unlike MAML, for each task, additional adversarial samples are generated and used to enhance the robustness for meta-training. Note that our algorithm is not restricted to any particular attack mechanism. Here, we use the *Fast Gradient Sign Method* (FGSM) proposed by Goodfellow et al. (2015) for illustration. For task $\mathcal{T}_i$, given a clean sample $(\mathbf{x}_c, \mathbf{y}_c)$ from $\mathcal{D}_i$, its corresponding adversarial sample $(\mathbf{x}_{adv}, \mathbf{y}_{adv})$ is generated using the following equations:

$$\begin{aligned} \mathbf{x}_{adv} &= \mathbf{x}_c + \epsilon \text{sign}(\nabla_{\mathbf{x}_c} J(f_{\boldsymbol{\theta}_{pre}}, \mathbf{x}_c, \mathbf{y}_c)); \\ \mathbf{y}_{adv} &= \mathbf{y}_c. \end{aligned} \tag{1}$$

where $J(f_{\boldsymbol{\theta}_{pre}}, \mathbf{x}_c, \mathbf{y}_c)$ represents the cost used to train a classification model $f_{\boldsymbol{\theta}_{pre}}$ parameterized by $\boldsymbol{\theta}_{pre}$, and $\epsilon$ specifies the size of the adversarial perturbation (the larger the $\epsilon$, the higher the perturbation). Note that the classification model $f_{\boldsymbol{\theta}_{pre}}$ is pre-trained based on the corresponding dataset and its parameters $\boldsymbol{\theta}_{pre}$ are fixed during the meta-training (Algorithm 1) and meta-testing.

The key idea behind ADML is to utilize antagonistic correlations between clean and adversarial samples to let the inner gradient update and the meta-update arm-wrestle with each other to obtain a good initialization of model parameters $\boldsymbol{\theta}$, which is robust to adversarial samples. Specifically, in the inner gradient update, we compute the new model parameters (updated in two directions) $\boldsymbol{\theta}_{adv_i}'$ and $\boldsymbol{\theta}_{c_i}'$ based on generated adversarial samples $\mathbf{D}_{adv_i}$, and clean samples $\mathbf{D}_{c_i}$ in training set $\mathbf{D}_i$ of task $\mathcal{T}_i$ respectively using gradient descent (Line 10). In the meta-update process, we update the model parameters $\boldsymbol{\theta}$ by optimizing the losses $\mathcal{L}_i(f_{\boldsymbol{\theta}_{adv_i}'})$ and $\mathcal{L}_i(f_{\boldsymbol{\theta}_{c_i}'})$ of the model with updated

---

**Algorithm 1** Adversarial Meta-Learner (ADML)

---

1: **Require:** $\alpha_1/\alpha_2$ and $\beta_1/\beta_2$: The step sizes for inner gradient update and meta-update respectively
2: **Require:** $\mathcal{D}$: The datasets for meta-training
3: **Require:** $< \mathcal{L}_i(\cdot) >$: The loss function for task $\mathcal{T}_i, \forall i \in \{1, \cdots, T\}$
4: Randomly initialize $\boldsymbol{\theta}$;
5: **while** not done **do**
6:     Sample batch of tasks $< \mathcal{T}_i >$ from task set $\mathcal{T}$;
7:     **for all** $\mathcal{T}_i$ **do**
8:         Sample $K$ clean samples $\left\{(\mathbf{x}_c{}^1, \mathbf{y}_c{}^1), ..., (\mathbf{x}_c{}^K, \mathbf{y}_c{}^K)\right\}$ from $\mathbf{D}_i$;
9:         Generate $K$ adversarial samples $\left\{(\mathbf{x}_{adv}{}^1, \mathbf{y}_{adv}{}^1), ..., (\mathbf{x}_{adv}{}^K, \mathbf{y}_{adv}{}^K)\right\}$ based on another $K$ clean samples from $\mathbf{D}_i$ to form a dataset $\{\mathbf{D}_{adv_i}, \mathbf{D}_{c_i}\}$ for the inner gradient update, containing $K$ adversarial samples and $K$ clean samples;
10:         Compute updated model parameters with gradient descent respectively:
        $\boldsymbol{\theta}'_{adv_i} := \boldsymbol{\theta} - \alpha_1 \nabla_{\boldsymbol{\theta}} \mathcal{L}_i(f_{\boldsymbol{\theta}}, \mathbf{D}_{adv_i}); \boldsymbol{\theta}'_{c_i} := \boldsymbol{\theta} - \alpha_2 \nabla_{\boldsymbol{\theta}} \mathcal{L}_i(f_{\boldsymbol{\theta}}, \mathbf{D}_{c_i});$
11:         Sample $k$ clean samples $\left\{(\mathbf{x}_c{}^1, \mathbf{y}_c{}^1), ..., (\mathbf{x}_c{}^k, \mathbf{y}_c{}^k)\right\}$ from $\mathbf{D}'_i$;
12:         Generate $k$ adversarial samples $\left\{(\mathbf{x}_{adv}{}^1, \mathbf{y}_{adv}{}^1), ..., (\mathbf{x}_{adv}{}^k, \mathbf{y}_{adv}{}^k)\right\}$ based on another $k$ clean samples from $\mathbf{D}'_i$ to form a dataset $\left\{\mathbf{D}'_{adv_i}, \mathbf{D}'_{c_i}\right\}$ for the meta-update, containing $k$ adversarial samples and $k$ clean samples;
13:     **end for**
14:     Update $\boldsymbol{\theta} := \boldsymbol{\theta} - \beta_1 \nabla_{\boldsymbol{\theta}} \sum_{\mathcal{T}_i \sim \mathcal{T}} \mathcal{L}_i(f_{\boldsymbol{\theta}'_{adv_i}}, \mathbf{D}'_{c_i}); \boldsymbol{\theta} := \boldsymbol{\theta} - \beta_2 \nabla_{\boldsymbol{\theta}} \sum_{\mathcal{T}_i \sim \mathcal{T}} \mathcal{L}_i(f_{\boldsymbol{\theta}'_{c_i}}, \mathbf{D}'_{adv_i});$
15: **end while**

---

parameters $\boldsymbol{\theta}'_{adv_i}$ and $\boldsymbol{\theta}'_{c_i}$ with respect to $\boldsymbol{\theta}$ based on the clean samples $\mathbf{D}'_{c_i}$ in testing set $\mathbf{D}'_i$ of task $\mathcal{T}_i$ and the corresponding adversarial samples $\mathbf{D}'_{adv_i}$ respectively:

$$
\min_{\boldsymbol{\theta}} \sum_{\mathcal{T}_i \sim \mathcal{T}} \mathcal{L}_i(f_{\boldsymbol{\theta}'_{adv_i}}, \mathbf{D}'_{c_i}) = \min_{\boldsymbol{\theta}} \sum_{\mathcal{T}_i \sim \mathcal{T}} \mathcal{L}_i(f_{\boldsymbol{\theta} - \alpha_1 \nabla_{\boldsymbol{\theta}} \mathcal{L}_i(f_{\boldsymbol{\theta}}, \mathbf{D}_{adv_i})}, \mathbf{D}'_{c_i});
$$
$$
\min_{\boldsymbol{\theta}} \sum_{\mathcal{T}_i \sim \mathcal{T}} \mathcal{L}_i(f_{\boldsymbol{\theta}'_{c_i}}, \mathbf{D}'_{adv_i}) = \min_{\boldsymbol{\theta}} \sum_{\mathcal{T}_i \sim \mathcal{T}} \mathcal{L}_i(f_{\boldsymbol{\theta} - \alpha_2 \nabla_{\boldsymbol{\theta}} \mathcal{L}_i(f_{\boldsymbol{\theta}}, \mathbf{D}_{c_i})}, \mathbf{D}'_{adv_i}).
$$

(2)

Note that in the meta-update, $\boldsymbol{\theta}$ is optimized in an *adversarial* manner: the gradient of the loss of the model with $\boldsymbol{\theta}'_{adv_i}$ (updated using adversarial samples $\mathbf{D}_{adv_i}$) is calculated based on clean samples $\mathbf{D}'_{c_i}$, while the gradient of the loss of the model with $\boldsymbol{\theta}'_{c_i}$ (updated using $\mathbf{D}_{c_i}$) is calculated based on adversarial samples $\mathbf{D}'_{adv_i}$. The arm-wrestling between the inner gradient update and the meta-update brings an obvious benefit: the model adapted to adversarial samples (through the inner gradient update using adversarial samples) is made suitable also for clean samples through the optimization of $\boldsymbol{\theta}$ in the meta-update based on the clean samples, and vice versa. So "*adversarial*" in ADML refers to not only adversarial samples but also the way of meta-training. *Please note that in Line 9, the set of adversarial samples $\boldsymbol{D}_{adv_i}$ can be generated exactly from the set of clean samples $\boldsymbol{D}_{c_i}$, and it also holds for $\boldsymbol{D}'_{adv_i}$ and $\boldsymbol{D}'_{c_i}$ in Line 12.*

The meta-update of the model parameters $\boldsymbol{\theta}$ is performed as the last step of each episode (Line 14). Through the arm-wrestling between the inner gradient update and the meta-update in the meta-training phase, $\boldsymbol{\theta}$ will be updated to a certain point, such that the average loss given by both adversarial samples and clean samples of all the tasks is minimized. In addition, we set the step sizes $\alpha_1 = \alpha_2 = 0.01, \beta_1 = \beta_2 = 0.001$, and set $\mathcal{L}_i(\cdot)$ of each classification task $\mathcal{T}_i$ to be the cross-entropy loss. $K$ and $k$ are task-specific, whose settings are discussed in the next section. Note that ADML preserves the model-agnostic property of MAML because both the inner gradient update and the meta-update processes are fully compatible with any learning model that can be trained by gradient descent.

We further illustrate the design philosophy of our algorithm in Figure 1. For each task $\mathcal{T}_i$, in the inner gradient update, ADML first drags $\boldsymbol{\theta}$ via gradient descent to the direction of the subspace that is favorable for adversarial samples (marked with the purple color) as well as another subspace that

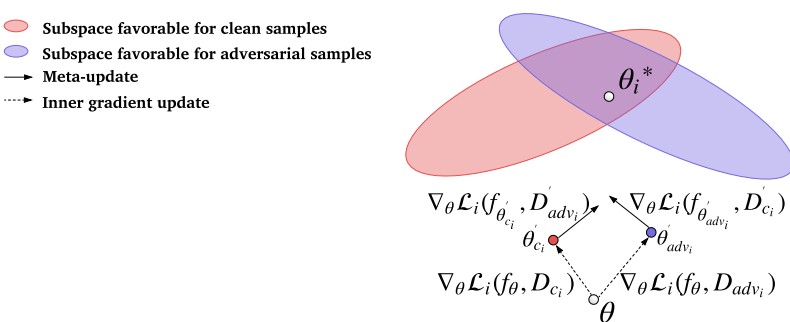

Figure 1: Illustration of design philosophy of ADML

is favorable for clean samples (marked with the red color) to reach two points $\boldsymbol{\theta}'_{adv_i}$ and $\boldsymbol{\theta}'_{c_i}$ respectively (i.e., Line 10). Then in the meta-update, based on $\boldsymbol{\theta}'_{adv_i}$ and $\boldsymbol{\theta}'_{c_i}$, ADML further optimizes $\boldsymbol{\theta}$ to its antithetic subspaces respectively (i.e., Line 14), and hopefully $\boldsymbol{\theta}$ can reach the optimal point $\boldsymbol{\theta}^*_i$, which is supposed to fall into the intersection of the subspace pair and is able to achieve a good trade-off between clean and adversarial samples to boost the overall performance on both samples. Note that here we only show the updates via a single task. Using all the tasks in $\mathcal{T}$, $\boldsymbol{\theta}$ can be optimized to a point with the smallest average distance to the intersections of all the subspace pairs, and thus can be quickly adapted to new tasks even with adversarial samples.

As mentioned before, a rather straightforward solution to the above adversarial meta-learning problem is to simply combine a meta-learner (e.g., MAML Finn et al. (2017)) with adversarial training (e.g., Goodfellow et al. (2015)). Specifically, we mix adversarial and clean samples to form both $\mathbf{D}_i$ (used in the inner gradient update) and $\mathbf{D}'_i$ (used in the meta-update), which are then used to calculate $\boldsymbol{\theta}'_i$ and update $\boldsymbol{\theta}$ using the following equations (just like MAML) respectively:

$$\boldsymbol{\theta}'_i = \boldsymbol{\theta} - \alpha\nabla_{\boldsymbol{\theta}}\mathcal{L}_i(f_{\boldsymbol{\theta}}, \mathbf{D}_i);$$
$$\boldsymbol{\theta} \leftarrow \boldsymbol{\theta} - \beta\nabla_{\boldsymbol{\theta}} \sum_{\mathcal{T}_i \sim \mathcal{T}} \mathcal{L}_i(f_{\boldsymbol{\theta}'_i}, \mathbf{D}'_i). \tag{3}$$

We call this method *MAML-AD*, which is used as a baseline for performance evaluation. However, it has been shown by our experimental results that although MAML-AD can slightly mitigate the problem, it still suffers from a significant performance degradation for new tasks with adversarial samples. This clearly shows that simply involving adversarial samples during the meta-training does not necessarily enhance the model's robustness; and well justifies that our idea of doing the inner gradient update and the meta-update in an adversarial way is necessary.

## 4 PERFORMANCE EVALUATION

The goal of our evaluation is to test and verify three properties of ADML: 1) ADML can learn quickly from limited data via a few gradient updates for a new task, and it is effective even in the cases with only clean samples; 2) ADML suffers from a minor performance degradation and yields much better performance than other meta-learning algorithms when encountering adversarial samples; 3) ADML works with adversarial samples generated by different attack mechanisms; and 4) ADML maintains stable performance when the perturbation of adversarial samples (i.e., $\epsilon$) escalates. In this section, we first introduce the experimental setup, and then present and analyze the results.

### 4.1 EXPERIMENTAL SETUP

In our experiments, we employed two commonly-used image benchmarks, MiniImageNet (Vinyals et al. (2016)), and CIFAR100 (Krizhevsky et al. (2009)). MiniImageNet is a benchmark for few-shot learning, which includes 100 classes and each of them has 600 samples. CIFAR100 was created originally for object recognition tasks, whose data are suitable for meta-learning, and same as MiniImageNet, it has 100 classes, each of which contains 600 images. Both MiniImageNet and

CIFAR100 were divided into 64, 16 and 20 classes for training, validation and testing respectively. Here we consider 5-way 1-shot and 5-way 5-shot classification tasks. 5 samples per class were used for the inner gradient update during meta-training of a 5-shot learning model (1 sample for 1-shot learning model). Thus, $K$ in ADML was set to 25 for 5-shot learning and 5 for 1-shot learning. 15 samples per class were used for the meta-update, thus we set $k = 75$. During the meta-testing, the learning model was trained on samples of 5 unseen classes, then tested by classifying new instances into these 5 classes.

Attack mechanisms including FGSM (Goodfellow et al. (2015)), FFGSM (Wong et al. (2020)), RFGSM (Tramèr et al. (2017)) and RPGD (Madry et al. (2017)) were leveraged to generate adversarial samples in the experiments. A well-trained VGG16 network (pre-trained on ImageNet and CIFAR100 respectively) for image classification served as the fixed classification model. To validate the performance of ADML with both strongly-attacked and slightly-attacked samples, the parameter $\epsilon$ was set to a large value (i.e., 2) and a small value (i.e., 0.2) for FGSM for both meta-training and meta-testing. See values of $\epsilon$ set for the other three attack mechanisms in supplementary material.

ADML was compared with four representative meta-learning algorithms, including MAML (Finn et al. (2017)), Matching Networks (Vinyals et al. (2016)), Relation Networks (Sung et al. (2018)) and R2D2 (Bertinetto et al. (2019)). Moreover, for fair comparisons, we compared ADML with another adversarial meta-learner MAML-AD (introduced in Section 3), which can be considered as a rather straightforward extension of MAML.

## 4.2 EXPERIMENTAL RESULTS

To fully test the effectiveness of ADML, we conducted a comprehensive empirical study, which covers various possible cases. The experimental results on MiniImageNet and CIFAR100 with FGSM attack mechanism are presented in Tables 1–2 and Tables 3–4 (in supplementary material) respectively. Experimental results on MiniImageNet with the other three attack mechanisms (i.e., FFGSM, RFGSM and RPGD) are shown in Tables 5–10 (in supplementary material). Each entry in these tables gives the average classification accuracy (with $95\%$ confidence intervals) of the corresponding test case, and the best results for each test case are marked in bold.

The experiments were conducted in six different test cases (combinations): "*Clean-Clean*", "*Clean-Adversarial*", "*Adversarial-Clean*", "*Adversarial-Adversarial*", "*40%-Clean*" and "*40%-Adversarial*". The first part of each combination (corresponding to a row) represents the training data used in the inner gradient update (or support set for Matching Networks and Relation Networks) during the meta-testing, while the second part (corresponding to a column) represents the testing data (or query set for Matching Networks and Relation Networks) for evaluation. "Clean" means clean samples only; "Adversarial" means adversarial samples only; and "40%" means that 40% samples of each class are adversarial and the rest 60% are clean, which represents intermediate cases. Note that the combinations, "40%-Clean" and "40%-Adversarial", do not exist for 1-shot learning since there is only one sample per class. Based on the results in Tables 1–2, we can make the following observations:

1) Just like MAML, ADML is indeed an effective meta-learner since it leads to quick learning from a small amount of new data for a new task. In the "Clean-Clean" cases, the general condition of meta-learning, ADML delivers desirable results, which are very close to the state-of-the-art given by R2D2, Relation Networks and MAML, and consistently better than those of MAML-AD and Matching Networks. For example, in the case of 5-way 1-shot classification with $\epsilon = 2$ (Table 1), ADML gives an average classification accuracy of 48.00% which is very close to that given by R2D2 (i.e., 49.52% and 51.80%), Relation Networks (49.67%) and MAML (48.47%) and it performs better than MAML-AD (43.13%) and Matching Networks (43.87%). *Note that the proposed ADML focuses on the cases with adversarial samples, and it is reasonable that it performs slightly worse on the cases with only clean samples.*

2) ADML is robust to adversarial samples since it only suffers from a minor performance degradation when encountering adversarial samples. For example, for the 5-way 5-shot classification with $\epsilon = 2$ (Table 2), ADML gives classification accuracies of 57.03%, 58.06%, 55.27%, 58.12% and 55.22% in the five test cases respectively. Compared to the "Clean-Clean" case (i.e., 59.38%), the performance degradation is only 4.16% in the worst-case and 2.64% on average. However, the classification accuracies given by the other meta-learning algorithms, including MAML-AD, drop

Table 1: Average classification accuracies on MiniImageNet with FGSM Attack (5-way, 1-shot)

| Method | Backbone | Meta-testing | $\epsilon = 2$ | | $\epsilon = 0.2$ | |
|---|---|---|---|---|---|---|
| | | | Clean | Adversarial | Clean | Adversarial |
| MAML | 32-32-32-32 | Clean | 48.47±1.78% | 28.63±1.54% | 48.47±1.78% | 42.13±1.75% |
| | | Adversarial | 28.93±1.62% | 30.73±1.66% | 42.23±1.85% | 40.17±1.76% |
| MAML-AD | 32-32-32-32 | Clean | 43.13±1.88% | 32.33±1.74% | 43.13±1.88% | 36.80±1.76% |
| | | Adversarial | 32.47±1.60% | 37.87±1.74% | 37.63±1.64% | 37.13±1.75% |
| Matching Nets | 64-64-64-64 | Clean | 43.87±0.41% | 30.02±0.39% | 43.88±0.48% | 36.14±0.40% |
| | | Adversarial | 30.45±0.44% | 30.80±0.43% | 36.58±0.49% | 35.03±0.39% |
| Relation Nets | 64-96-128-256 | Clean | 49.67±0.85% | 32.32±0.58% | 49.45±0.84% | 43.03±0.74% |
| | | Adversarial | 32.59±0.79% | 32.85±0.63% | 42.98±0.85% | 40.89±0.79% |
| R2D2 (64C) | 64-64-64-64 | Clean | 49.52±1.70% | 25.80±1.06% | 49.52±1.70% | 41.24±1.41% |
| | | Adversarial | 28.35±1.57% | 30.94±1.19% | 40.94±1.67% | 38.74±1.39% |
| R2D2 | 96-192-384-512 | Clean | **51.80±1.70%** | 25.95±0.88% | **51.80±1.70%** | 42.74±1.30% |
| | | Adversarial | 29.66±1.63% | 31.39±1.14% | 43.27±1.81% | 39.36±1.57% |
| ADML (Ours) | 32-32-32-32 | Clean | 48.00±1.87% | **43.00±1.88%** | 48.00±1.87% | **43.20±1.70%** |
| | | Adversarial | **40.10±1.73%** | **40.70±1.74%** | **44.00±1.83%** | **41.20±1.75%** |

Table 2: Average classification accuracies on MiniImageNet with FGSM Attack (5-way, 5-shot)

| Method | Backbone | Meta-testing | $\epsilon = 2$ | | $\epsilon = 0.2$ | |
|---|---|---|---|---|---|---|
| | | | Clean | Adversarial | Clean | Adversarial |
| MAML | 32-32-32-32 | Clean | 61.45±0.91% | 36.65±0.88% | 61.47±0.91% | 53.05±0.86% |
| | | 40% | 56.74±0.93% | 43.05±0.86% | 59.25±0.91% | 54.67±0.91% |
| | | Adversarial | 41.49±0.95% | 45.46±0.97% | 55.19±0.95% | 53.33±0.92% |
| MAML-AD | 32-32-32-32 | Clean | 57.13±0.96% | 41.65±0.92% | 57.09±0.96% | 49.71±0.88% |
| | | 40% | 54.07±0.91% | 48.74±0.91% | 56.52±0.90% | 52.08±0.90% |
| | | Adversarial | 43.21±0.91% | 52.07±0.96% | 51.23±0.90% | 51.36±0.94% |
| Matching Nets | 64-64-64-64 | Clean | 55.99±0.47% | 33.73±0.39% | 55.55±0.44% | 44.91±0.40% |
| | | 40% | 49.88±0.45% | 35.67±0.44% | 52.72±0.45% | 45.65±0.42% |
| | | Adversarial | 36.24±0.45% | 37.91±0.40% | 47.77±0.45% | 46.19±0.44% |
| Relation Nets | 64-96-128-256 | Clean | 63.85±0.73% | 38.37±0.64% | 63.86±0.73% | 55.39±0.68% |
| | | 40% | 56.53±0.77% | 41.04±0.67% | 59.02±0.70% | 55.06±0.68% |
| | | Adversarial | 42.74±0.79% | 46.08±0.63% | 56.85±0.73% | 53.65±0.69% |
| R2D2 (64C) | 64-64-64-64 | Clean | 65.48±1.35% | 29.34±1.22% | 65.48±1.35% | 51.60±1.24% |
| | | 40% | 59.28±1.65% | 40.33±1.20% | 62.08±1.49% | 54.56±1.29% |
| | | Adversarial | 36.07±1.59% | 43.00±1.38% | 55.27±1.61% | 53.97±1.32% |
| R2D2 | 96-192-384-512 | Clean | **68.42±1.28%** | 29.95±1.11% | **68.42±1.28%** | 54.60±1.31% |
| | | 40% | **61.11±1.56%** | 40.38±1.35% | **64.27±1.46%** | 56.93±1.36% |
| | | Adversarial | 37.69±1.52% | 44.07±1.36% | 58.53±1.68% | 55.71±1.39% |
| ADML (Ours) | 32-32-32-32 | Clean | 59.38±0.99% | **57.03±0.98%** | 61.28±0.99% | **58.33±0.96%** |
| | | 40% | 58.12±0.90% | 55.22±0.98% | 60.09±0.89% | **59.13±0.92%** |
| | | Adversarial | **58.06±0.96%** | **55.27±0.92%** | 59.02±0.88% | **57.74±0.93%** |

substantially when there are adversarial samples. For example, for MAML, the accuracy drops from 61.45% ("Clean-Clean") to 36.65% ("Clean-Adversarial") when injecting adversarial samples for testing, which represents a significant degradation of 24.80%. Similar observations can be made for MAML-AD, Matching Networks, Relation Networks, R2D2 (64C) and R2D2, which represent substantial degradations of 15.48%, 22.26%, 25.48%, 36.14% and 38.47% respectively.

3) ADML generally outperforms all the other meta-learning algorithms in the test cases with adversarial samples. For instance, in the "Clean-Adversarial" cases of 5-way 5-shot learning with $\epsilon = 2$ (Table 2), ADML achieves an accuracy of 57.03%, which represents 20.38%, 15.38%, 23.30%, 18.66%, 27.69% and 27.08% improvements over MAML, MAML-AD, Matching Networks, Relation Networks, R2D2 (64C) and R2D2 respectively. This clearly shows the superiority of the adversarial meta-training procedure of the proposed ADML compared to MAML-AD, the straightforward adversarial meta-learner. Even so, MAML-AD still generally performs better than the rest meta-training algorithms, when dealing with adversarial samples. Note that an exception occurs in the case "40%-Clean", where the accuracy of ADML is slightly lower than that of R2D2, even with 40% adversarial samples. This implies that in the 5-shot learning cases with adversarial samples,

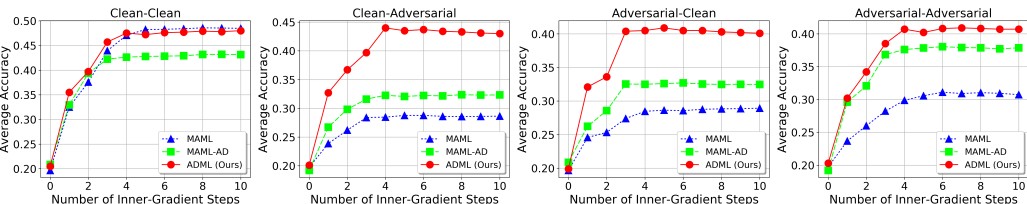

Figure 2: Average loss over the gradient update step for 5-way 1-shot learning on MiniImageNet

Figure 3: Top-1 accuracy over the gradient update step for 5-way 1-shot learning on MiniImageNet

the clean samples (even with a number smaller than 5) in the training data play an important role when classifying the testing data composed of clean samples.

4) When the perturbation of adversarial samples escalates, ADML maintains stable performance. For example, for 5-way 1-shot learning, when $\epsilon$ increases from $0.2$ to $2$ (Table 1), ADML only leads to minor degradations of $0.2\%$, $3.9\%$ and $0.5\%$ in the corresponding three cases involving adversarial samples. However, much more significant degradations can be observed when the other meta-learning algorithms are applied. For instance, the accuracies of MAML suffer from $13.50\%$, $13.30\%$ and $9.44\%$ drops in these three cases when increasing $\epsilon$ from $0.2$ to $2$.

5) As expected, when lower perturbations exerted, the adversarial samples are not significantly different from the corresponding clean samples, which brings about relatively close results in different test cases, not only for ADML, but also for the other meta-learning algorithms. For instance, for the 5-way 5-shot classification with $\epsilon = 0.2$ (Table 2), MAML gives a smaller gap of $8.42\%$ between "Clean-Clean" and "Clean-Adversarial", compared with that of $24.80\%$ when $\epsilon = 2$.

*Similar observations can be made for the results corresponding to CIFAR100 and the other three attack mechanisms ((i.e., Tables 3–10 in supplementary materials)).* In addition, we show how the loss and Top-1 accuracy change in Figures 2 and 3 during the meta-testing. Specifically, these two figures show that when ADML, MAML and MAML-AD are applied on MiniImageNet, how the losses and Top-1 accuracies change with the gradient update step during the meta-testing in the four test cases of 5-way 1-shot learning with $\epsilon = 2$. We observe that, for all the cases, the losses of the models learned with ADML drop sharply after only several gradient updates, and stabilize at small values during the meta-testing, which are generally lower than those of the other two methods. Moreover, the Top-1 accuracies of the models learned with ADML rise sharply after only several gradient updates, and stabilize at values, which are generally higher than those of the other two methods (a little bit lower than that of MAML in "Clean-Clean" case). *Similar trends can be observed in Figures 4–5 in supplementary materials for 5-way 5-shot learning with $\epsilon = 2$.* These observations further confirm that ADML is suitable for meta-learning since it can quickly learn and adapt from small data for a new task through only a few gradient updates.

## 5 CONCLUSIONS

In this paper, we proposed a novel method called ADML (ADversarial Meta-Learner) for meta-learning with adversarial samples, which features an *adversarial* way for optimizing model parameters $\boldsymbol{\theta}$ during meta-training through the arm-wrestling between inner gradient update and meta-update using both clean and adversarial samples. The extensive experimental results have showed

that 1) ADML is an effective meta-learner even in the cases with only clean samples; 2) a straightforward adversarial meta-learner, namely, MAML-AD, does not work well with adversarial samples; in addition, 3) ADML is robust to adversarial samples generated by different attack mechanisms, and outperforms other meta-learning algorithms including MAML on adversarial meta-learning tasks; and most importantly, 4) it opens up an interesting research direction and sheds light on dealing with the difficult cases with limited and even contaminated samples.

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
