# OpenReview forum: "Adversarial Meta-Learning"
_ICLR.cc/2021/Conference — Reject_

### Official Review · AnonReviewer2 · 2020-10-28
**This paper presents a novel ADversarial Meta-Learner (ADML), the claimed first work that tackles adversarial samples in meta-learning. In its core algorithm, a given model is diverged simultaneously by a set of clean  samples and a set of perturbed counterparts, to generate two inner models, which are then meta-updated in an adversarial manner using new sets of clean/adversarial samples.**

**Rating:** 5
**Confidence:** 4

**Review:**

Summarization of the contribution:

This paper presents a novel ADversarial Meta-Learner (ADML), the claimed first work that tackles adversarial samples in meta-learning. In its core algorithm, a given model is diverged simultaneously by a set of clean  samples and a set of perturbed counterparts to generate two inner models, which are then meta-updated in an adversarial manner using new sets of clean/adversarial samples.


Strengths:

The paper addressed an important topic in meta-learning, i.e., meta-learning with adversarial samples. On two datasets, the experiments, especially the ablation studies (page 6), are carefully designed, and the results were also discussed
 in detail (page 7-8).


Weaknesses:

1. The paper gives no sensitivity analysis of the hyper-parameters. For example, on page 4, \alphas and \betas were set at different values but no reason was given.

2. There is neither open-sourced code nor plan for releasing the code in the paper. Thus, it is questionable whether the proposed method is reproducible.

3. The idea of combining meta-learning with adversarial defense has already been proposed and well studies, such as in "Adversarial Attacks on Graph Neural Networks via Meta Learning, ICLR 2019".

4. The experiments are not thoroughly conducted. MiniImageNet and CIFAR100 are all small scale datasets with only 60k samples each. Indeed, image classification is an important task and a benchmark for meta-learning. Still, it would be more informative to test the proposed method with multiple types of data, say graph data as mentioned above.

5. Meta-learning, by its definition, is a learning framework, which is not necessarily tantamount to few-shot/one-shot learning, and is different from the latter in many ways. However, the whole experiment section has focused on few-shot learning using classic few-shot learning tasks. This gives us an impression that the title "adversarial meta learning" is somewhat misleading. Actually, a recent work "Adversarially Robust Few-Shot Learning: A Meta-Learning Approach, NeurIPS 2020" adopted
 nearly the same setting and datasets and achieved superior results.

6. Another main problem is that there was, unfortunately, no theoretic analysis on the proposed algorithm, but only some intuition behind its design. It would be interesting to dive deeper, mathematically, to show how the algorithm
 works under the hood and to be more confident about where and when it holds true.


Recommendation and reason: Weak Reject.

Overall, this is a good paper.  However, it is a little bit below the standard of ICLR, considering its novelty, experiments, writing, and depth of analysis.


Additional comments:

1. The language of the paper needs to be polished and refined.
2. Page 1: it is better to rephrase the sentence about MAML to give a clearer description of it.
3. Page 3: each task is a 5-way classification task: it is better to explain what is a "5-way" task for the readers who are not familiar with few-shot learning.
4. Page 6: the paper considered several attack mechanisms including FGSM, FFGSM, RFGSM, and RPGD. But in reality, the attacker may not divulge the mechanism to be used. So can this method be attack mechanism-agnostic
 i.e. be resistant to an attack without having known the mechanism in advance?

---

### Official Review · AnonReviewer4 · 2020-10-28
**A simple yet effective approach to extend MAML against adversarial attacks**

**Rating:** 6
**Confidence:** 4

**Review:**

This paper presents ADML (ADversarial Meta-Learner), a method for general meta-learning when adversarial samples are present. In a sense, ADML extends MAML to deal with adversarial/contaminated samples in training. Building on top of MAML, the key insight is to validate or ground the model's updates from both angles: 1) to ground the parameter update from clean examples onto an adversarial set; 2) to ground the parameter update from the adversarial examples onto a clean set. Experimental evaluations on MiniImageNet and CIFAR 100 demonstrates the good performances of ADML over a series of meta-learning baselines. In addition, it's also shown that simply creating a mixed dataset with both clean and adversarial examples (i.e., MAML-AD) does not fully address the adversarial signals, thus validating the key insights of the two-way "cross-grounding" technique.

The paper is well written and the extension of MAML onto the adversarial setting is simple yet effective. One minor comment regarding the evaluation though is that there could a couple more ablation studies for ADML. For example, how sensitive is the ADML to the learning rates for the outer objective? Also for the clean-clean case, since MAML still performs slightly better than ADML, is it possible to modify/extend ADML such that it will still match MAML's performance when there's no adversarial examples in the training set?

---

### Official Review · AnonReviewer3 · 2020-10-29
**A nice proposal, somewhat lacking in depth**

**Rating:** 4
**Confidence:** 4

**Review:**

Summary

This paper presents a variant of MAML that aims to make the meta-learned initialisation robust to adversarial examples. The core idea of the proposed method, ADML, is to compute two task-specific parameters in the inner-loop: one on clean data and one on adversarial data. In the meta-learning step, the clean task-specific parameters are meta-learned on adversarial data while the adversarial task-specific parameters are meta-learned on clean data. The authors demonstrate robustness to a range of adversarial attacks on miniImagenet.

Pros:
- As far as I'm aware, this is the first proposal to meta-learn for adversarial robustness
- Method is agnostic to the model and the form of adversarial attack
- Reported results demonstrate robustness to a range of adversarial attacks

Cons:
- The method is not really explored in any depth
- The method is limited to MAML
- Considered attacks are relatively weak [see 1]

Recommendation: rejection

Motivation:

While the idea of adversarial meta-training is well motivated and generally sound, the specific method in this paper is primarily proposed and not really explored in any depth. The authors demonstrate that a simple approach of mixing adversarial and clean data in the usual MAML update doesn't work very well, but don't go in to any details as to why this failure motivates their method. In particular, ADML formulates *two* task optimisation problems (Eq. 2) for the same variable (the initialisation) - one is adaptation under clean data and one under adversarial data. While intriguing, I'm missing a motivation: why would we want to consider two independent adaptation trajectories under best / worst case scenario, and not one trajectory under mixed data?

Given that ADML formalises two independent optimisation problems over the same variable, it is not at all clear how they should be combined. The algorithm itself does not indicate how this is done, the text indicates that the losses are simply averaged? If so, that induces a specific optimisation problem that is neither of the two presented in Eq. 2. Have you considered what what that optimisation problem looks like?

Perhaps the least obvious choice in ADML is to swap clean and adversarial data in the meta-update. While interesting, there is no apparent explanation for this and as a reader I would like to know what motivates this design choice and how much it really matters. Given that the experimental section is restricted to miniImagenet only, some form of simple analysis would have gone a long way to shed light on the proposed method. I would have liked to see an ablation on the design choices of ADML to gain insight into which of these are responsible for the observed performance gain.

Finally I have a concern with the evaluation protocol: the algorithm suggests that adversarial examples are generated by drawing a fresh batch of data. In a K-shot regime, does this mean that you draw two batches of K samples per task adaptation? This would mean that ADML see twice as much data and result in unfair comparisons.

Minor issues
- There are several grammar and spelling mistakes that makes the paper look rushed (pages 1, 2, 3)
- In the motivation, the authors use the phrase 'arm-wrestle'. What is meant by this analogy?
- Problem statement: a meta-learner (at least MAML) does not take several datasets as input, only one at a time.
- why are adversarial examples generated from a different batch of task data? Could they not be generated from the same batch?
- Algorithm: (1) the variables \bar{D} are never used and (2) L14 makes a nonsensical double assignment to \theta.
- Figure 1: the meta-update arrows should originate in the initialisation since that is what they are updating.
- Experimental setup: you mention that 15 adversarial attack mechanisms where leveraged in the experiments, but the tables suggests that you only use 1 (at a time)?
- Why does MAML-AD do worse on clean-clean? Should it not be exactly equivalent to MAML in this case?

Typos:
- "none of existing works have well addressed" -> "existing works have not yet addressed"
- "which, however, are" -> ", which is"
- "an existing meta-learning algorithms" -> "[...] algorithm"
-  "we show such a approach" -> "we show such an approach"
- "testing dataset" -> "test dataset"
- "In the meta-training" -> "In the meta-training phase"

Post Rebuttal

I have read the authors rebuttal and updated manuscript. The authors have taken some steps to clarify parts of the manuscript, but my main concerns remain and thus my score is unchanged. In particular, the authors did not clarify how the two optimisation problems in Eq. 2. relates and what this means algorithmically. The authors defend their empirical setup by stating that meta-testing is still K-shot. While this is true, it is also true that their method have enjoy a greater amount of meta-training compared to baselines. What I would have liked to see is an ablation that trains the baselines for 2x meta-updates, but this is unfortunately not provided.

Echoing other reviewers, if stronger attacks are considered, these should not be relegated to appendix.

References
[1] Goldblum et. al. Adversarially Robust Few-Shot Learning: A
Meta-Learning Approach. 2020.

---

### Official Review · AnonReviewer1 · 2020-10-29
**Investigates a new problem in FSL, but not comprehensively**

**Rating:** 4
**Confidence:** 5

**Review:**

The paper focuses on few shot learning with adversarial training samples in both during meta-training and meta-testing phases. The paper builds their model on top of MAML with an addition of a unique training scheme that interchanges adversarial and clean examples of tasks used in inner gradient and outer gradient optimization of MAML. Instead of mixing up clean and adversarial examples in both inner gradient optimization and outer gradient optimization the paper suggests using them in a mutually complementary fashion. Here, the core idea is to estimate the target initial MAML model such that when the model updated with clean support examples is robust against adversarial query samples, and vice versa.

The paper is overall definitely interesting because adversarial attacks in few-shot learning is not a studied topic, to the best of my knowledge, and the proposed approach appears to be effective, especially compared to a meaningfully-defined baseline called MAML-AD.

However, the paper suffers from three main shortcomings. First, only FGSM is used as the adversarial sample generation method and other well-known techniques are not considered. It could have been much better to incorporate several other well-established attack techniques.

Second, the paper presumes that the same attack technique is being used during training & testing. This is hardly a realistic assumption. It could have been definitely interesting to see the robustness of the ADML model when it is attacked by a different model at test time (in generation of query and/or support samples of few-shot classes).

Third, some simple & directly applicable defense methods, such as local spatial smoothing, are not considered. Such techniques could have been used to define additional baselines.

To sum up:

Pros:
- Paper is well written, gives sufficient background information and states the ideas clearly,
- Defines & tackles a novel problem,
- Reports promising results.

Cons:
- Only FGSM is being evaluated as the adversarial example generation method.
- The same attack approach (FGSM) is used for both training and testing.
- Existing defense techniques are not utilized.
- Although intuitively & experimentally ADML is convincing compared to MAML-AD, the theoretical justification is weak. (This is a rather minor point as most state-of-the-art methods in FSL & adversarial attacks are explained only intuitively)
- It is unclear why (only) matching nets & relation nets are being considered as baselines, other (more) recent methods could have been evaluated for robustness,

Finally, minor points: there is a typo on P1 ("decent") and the works on adversarial attacks on related problems, such zero-shot learning, can be included into the discussion for completeness.

Post-rebuttal: I'd like to thank the authors for the rebuttal and for pointing out some additional results that I had initially missed. While they're definitely welcome, it is still not clear to me why the main work is based on FGSM and why simple defense techniques have not been considered, both of which significantly weaken the manuscript.  I also see that very similar concerns have been raised in the other reviews , confirming that these weaknesses of the manuscript are quite salient.

---

### Official Review · AnonReviewer5 · 2020-11-04
**Emergency review for "Adversarial Meta-Learning"**

**Rating:** 3
**Confidence:** 5

**Review:**

### Contributions ###
* The authors study a new kind of problem, the robustness of few-shot learners against adversarial attacks
* The paper propose a new meta-learning algorithm ADML that uses adversarial and clean examples during meta-training (both for train-train and train-test).

### Significance ###
Combining meta-learning with adversarial robustness is an interesting research direction that might inspire follow-up work. However, it remains unclear if any practical applications would benefit from adversarially robust few-shot learners. A clearer motivation would be desirable.

### Originality ###
In principle, combining gradient-based meta-learning such as MAML with adversarial training (similar to the proposed baseline MAML-AD) is a natural fit that does not require novel research. The novel methodological contribution of ADML is that meta-train-train is done separately for clean and adversarial examples and that meta-train-test evaluates the loss of the adversarially trained parameters on clean train-test data (and vice versa). While this is an interesting idea and empirical evidence seems to support this idea (see below for some major caveat), the paper lacks any theoretical justification for ADML. At least a well motivated intuition would be required.

### Clarity ###
Some details remain unclear:
* From Algorithm 1, it seems that 2K samples are used for meta-train-train and 2k for meta-train-test, because clean and adversarial examples are constructed based upon different samples. This would make it an uneven comparison to standard meta-learning, that would do actual K-shot training.
* For adversarial attacks like PGD, hyperparameters like step-size and number of steps are not stated

### Quality ###
The paper does not follow established procedures for evaluating adversarial robustness:
* the threat model is not clearly stated
* the main evaluation in the paper is done based upon a weak attack, namely FGSM
* most importantly, the authors are using only transfer attacks for robustness evaluation (and also during meta-training). That is: adversarial examples are computed based upon a pretrained network  (paragraph after Equation 1). This is not at all motivated; the natural thing would be to compute adversarial examples always based on the current model (\theta in Algorithm 1)

In summary, the presented results need to be taken with care because all observed effects might be due to using weak attacks.

### Recommendation ###
Because of the lack of motivation for adversarial meta-learning, lacking intuition for ADML, and particularly the robustness evaluation against a very weak attack, the paper is a clear reject to me.

### Recommendation after Author Response ###
I have read the authors response and appreciate the clarification by the authors. I was aware that other attacks (FFGSM, RFGSM, RPGD), were in the supplementary material. However, I don't see any reason for focusing on the particularly weak FGSM in the main paper. The main evaluation should always be based on the strongest not the weakest attack. Moreover, also RPGD was used as a transfer attack and as such is quite weak.
In summary, I keep my recommendation of rejecting the paper.

---

### Decision · Program_Chairs · 2021-01-07
**Final Decision**

**Decision:**

Reject

**Comment:**

The paper proposes a new meta-learning algorithm which promises greater robustness to adversarial examples. I will be brief, as the fault with the paper is quite clear: the experimental results are not sufficient. The attack used (FGSM) is particularly dated and weak, and the comparison to existing defences is insufficient. Additionally, prior work (Adversarially Robust Few-Shot Learning: A Meta-Learning Approach) obtains better results, and is not compared against. The reviewers provided further criticism regarding the motivation for and explanation of the method, but the empirical aspects of the paper are where it primarily falls short of the publishable standard for ICLR.

I recommend rejection, and invite the authors to consider demonstrating robustness to a wider range of attacks (including non-gradient based), and a more thorough comparison to defence methods, before resubmitting to another conference.